# Usual gait speed is inversely associated with depression in middle-aged and older adults: A cross-sectional study in Korea

Jae Ho Park[1], Joong-Yeon Lim[1], Hyun-Young Park[2]*

1 Division of Population Health Research, Department of Precision Medicine, National Institute of Health, Cheongju-si, Chungcheongbuk-do, Korea, 2 National Institute of Health, Cheongju-si, Chungcheongbuk-do, Korea

* mdhypark@gmail.com

## Abstract

### Background

Depression is a serious mental disorder and leading cause of suicide. This study investigated the association between usual gait speed (UGS) and risk of depression.

### Methods

Data from 2,419 participants from a community-based Korean cohort were analyzed. Participants were categorized into sex-specific UGS tertiles (low, mid, or high). Depression was defined based on a previous physician diagnosis, current use of antidepressants, or a score of ≥6 on the Korean version of the Geriatric Depression Scale-Short Form (SGDS-K). Multiple linear and logistic regression models were used to assess the association between UGS and SGDS-K scores and estimate the odds ratios (ORs) with 95% confidence intervals (CIs) for the risk of depression, respectively.

### Results

Prevalence rates of depression were 13.33% and 26.29% among men and women, respectively. Compared with participants with low UGS, men with high UGS had a 50% (OR=0.50; 95% CI [0.29, 0.86]; $p < 0.05$) lower risk of depression, and women with mid and high UGS had a 43% (OR=0.57; 95% CI [0.41, 0.79]; $p < 0.001$) and 44% (OR=0.56; 95% CI [0.38, 0.82]; $p < 0.01$) lower risk, respectively. The SGDS-K scores were lowered by 0.14 (95% CI [–0.23, –0.04]; $p < 0.01$) and 0.33 points (95% CI [–0.45, –0.21]; $p < 0.0001$) in men and women, respectively, with each 0.1 m/s increase in UGS.

**Data availability statement:** "No - some restrictions will apply; Data availability statement: The data used in this study were obtained from the Korean Genome and Epidemiology Study (KoGES; 6635-302) of the National Institute of Health, Korea. Details of the data request process and contact information can be accessed at: https://www.nih.go.kr/ko/main/contents.do?menuNo=300566".

**Funding:** This research was supported by the National Institute of Health (NIH) research project (Grant No. 2024-NI-003-01).

## Conclusions

Hence, faster UGS was significantly associated with a reduced risk of depression in both sexes. Thus, maintaining a fast UGS may have protective benefits against the risk of depression.

## Introduction

Depression, also known as depressive disorder, is a common and serious mental health condition. Meta-analytical evidence suggests that depression significantly increases the incidence risk of cardiovascular disease (CVD), liver and lung cancers, and even suicidal ideation and attempts [1–3]. Furthermore, a recent study estimated an additional 53.2 million global cases of depression in 2020 owing to the coronavirus disease 2019 (COVID-19) pandemic [4]. Similarly, its prevalence increased from 4.3% in 2018 to 5.2% in 2020 among adults in Korea due to the pandemic [5]. These trends underscore the increasing need for strategies to prevent and manage depression in the post-pandemic period.

Since walking is an essential part of daily life, walking ability is an important factor in independent life. Among the walking parameters, studies have utilized usual gait speed (UGS), walking speed at a usual pace, to assess overall functional fitness, physical performance, and sarcopenia [6,7]. Slower UGS is a clinical indicator of the future risk of mobility disability, cognitive impairment, dementia, CVD, and even mortality [8–10]. However, UGS decreases with age in adulthood [11] due to shorter step length resulting from reduced range of motion (ROM), balance ability, and muscular strength of the lower extremities [12–14]. Thus, maintaining and/or enhancing UGS is essential for preventing these life-threatening conditions.

Decline in physical function is also associated with mental health consequences. Among the physical function parameters, UGS is positively correlated with social participation; however, lower-extremity muscular strength and balance are not [15]. Moreover, UGS is negatively related to social isolation and loneliness [16]. Since social isolation, loneliness, and reduced social participation are closely associated with an increased risk of depression [17,18], interest in the relationship between UGS and depression is increasing. Several studies have reported that slower UGS is associated with an increased risk of depression in Western countries, such as Turkey [19], England [20], the Netherlands [21], and northern Sweden and western Finland [22]; however, they did not examine sex-based differences through stratified analyses. A study reported that UGS changes exhibit a gradual decline in men in adulthood and a marked decrease at menopause among women [11]; furthermore, previous studies have reported a significantly higher proportion of women in slow UGS groups. Thus, sex-disaggregated analyses are warranted. To our best knowledge, few epidemiological studies have focused on this relationship in Asian populations, especially among Koreans. Further research should verify the potential variations across sex, ethnicity, and race.

The Korean version of the Geriatric Depression Scale-Short Form (SGDS-K) is a reliable and valid instrument for diagnosing depression in the older adult Korean

population [23,24]. Furthermore, its score is inversely associated with well-being, quality of life, and cognitive function [25] and positively correlated with oxidative stress biomarkers, such as urinary malondialdehyde levels [26]. Taken together, these findings suggest that the SGDS-K score functions as a diagnostic instrument for depression and an independent measure directly associated with the key determinants of its development. Therefore, we aimed to investigate whether UGS was significantly associated with the risk of depression and SGDS-K score. To the best of our knowledge, only a few studies have investigated these associations.

Therefore, this study examined the association between UGS and the risk of depression in middle-aged and older Korean adults via a population-based cohort. Furthermore, we investigated the relationship between UGS and the SGDS-K scores.

## Materials and methods

### Study participants

This population-based cross-sectional study used data from the eighth wave (2015–2016) of the Anseong cohort, part of the Korean Genome and Epidemiology Study (KoGES) that aims to establish comprehensive healthcare guidelines for non-communicable diseases. The eighth wave of this ongoing prospective population-based cohort included 3,235 participants aged 53–84 years who resided in Anseong-si, Gyeonggi-do, Korea. We used this wave as it measured USG in the largest number of participants. Face-to-face surveys and physical examinations were conducted by trained medical staff. We obtained and analyzed the dataset provided by the National Institute of Health, Korea, on February 20, 2025. The KoGES ensures compliance with the Personal Information Protection Act and Statistics Act by only providing de-identified data. Thus, our access was limited to anonymized datasets devoid of personally identifiable information. Detailed information on this cohort has been previously reported [27].

Of the 3,235 participants, individuals were excluded if they had missing SGDS-K data (n = 10), missing UGS data (n = 383), clinical history of dementia (n = 27), or missing covariate data (n = 396). After exclusion, 2,419 participants (1,354 women) were analyzed (Fig 1). Written informed consent was obtained from all the participants. This study was approved by the Institutional Review Board of the National Institute of Health, Korea Disease Control and Prevention Agency (Approval No. KDCA-2024-02-12-R-04).

### Measurement of UGS

For the 4-meter walk test, trained healthcare providers asked participants to walk a 4-meter course from a starting to end point at their usual pace. The time (in seconds) it took for both feet to pass the end point was recorded with a digital stopwatch for two trials, and the faster time was used. UGS was calculated by dividing 4 m by the time in seconds (m/s). Participants were categorized into sex-specific UGS tertiles (low, mid, or high).

### Definition of depression

Depression was defined based on a previous diagnosis by a physician, current use of antidepressants, or a score of ≥6 points on the 15-item SGDS-K. The SGDS-K is a reliable and valid instrument for screening depression in the older adult Korean population [23,24]. According to a previous study, a score of 6 was the optimal cut-off point for screening both minor and major depressive disorders [28]. Participants were categorized into two groups, "non-depression" (those without depression) and "depression" (those with depression), for comparison.

### Covariates

Analysis included sociodemographic and health-related factors, such as age, sex, marital status, educational level, household income, drinking and smoking habits, regular exercise, body mass index (BMI), waist circumference, blood pressure

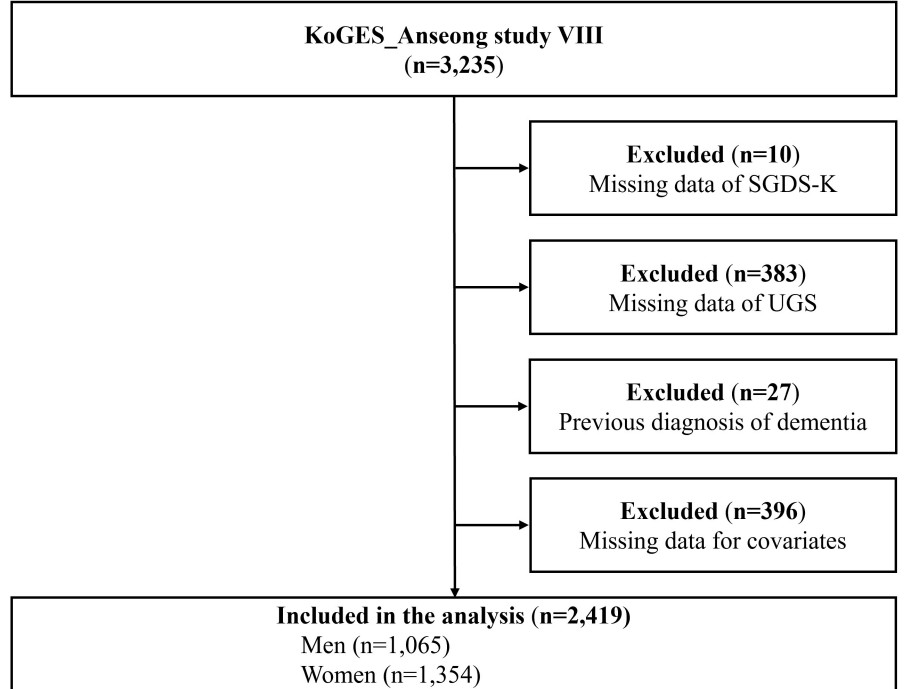

**Fig 1. Flow diagram of participants' inclusion and exclusion.** SGDS-K, Korean version of the Geriatric Depression Scale-Short Form; UGS, usual gait speed.

(BP), hypertension, diabetes mellitus, and laboratory parameters. Marital status was classified as "divorced/widowed/single" or "married/partnered." Educational level was categorized as elementary school or lower, middle or high school, or college or higher. Household income was classified as "<1," "1–<2," "2–<3," "3–<4," and "≥4" million KRW/month (million Korean won per month). Drinking and smoking habits were classified as "never," "former," or "current." Regular exercise was defined as engaged in ≥150 min/week of moderate-intensity exercise in a week. Moderate-intensity exercise was defined as participation in sports or exercises that induce sweating.

Anthropometric measurements, which included height, body weight, and waist circumference, were taken by trained healthcare providers via standardized protocols. BMI was calculated as body weight (kg) divided by height (m) squared (kg/m$^2$). Trained healthcare providers also measured participants' BP via standard methods. Systolic (SBP) and diastolic BP (DBP) were obtained by averaging two readings from the arm with the highest SBP after the participant rested for 5 min in a seated position. Blood samples were collected for biochemical assays after 8 hours of overnight fasting, and total cholesterol (T-Chol), high-density lipoprotein cholesterol (HDL-C), triglyceride (TG), and fasting blood glucose (FBG) levels were measured. Hypertension was identified based on a previous physician diagnosis, current use of antihypertensive drugs, SBP ≥ 140 mmHg, or DBP ≥ 90 mmHg. Diabetes mellitus was defined based on a previous physician diagnosis, current use of antidiabetic medications, which included insulin or oral hypoglycemic agents, FBG ≥ 126 mg/dL, glycated hemoglobin ≥6.5%, or 2-hour blood glucose ≥200 mg/dL in the oral glucose tolerance test. Detailed information on the biochemical analyses is available elsewhere [27].

## Statistical analysis

All statistical analyses were conducted using SAS software (version 9.4; SAS Institute, Cary, North Carolina, United States). Participants' characteristics were summarized via descriptive statistics, with continuous and categorical variables

presented as mean±standard deviation and absolute frequencies and percentages (%), respectively. Chi-squared tests were used to assess intergroup differences in sex, marital status, educational level, household income, drinking and smoking habits, regular exercise, and prevalence of hypertension, diabetes mellitus, and depression. Independent *t*-tests and one-way analysis of variance (ANOVA) were conducted to compare age, UGS, the SGDS-K score, BMI, waist circumference, SBP, DBP, T-Chol, HDL-C, TG, and FBG levels between the groups. Scheffé tests were used for post-hoc comparisons if the ANOVA revealed significant differences.

Multiple logistic regression models were used to evaluate the odds ratios (ORs) and 95% confidence intervals (CIs) of depression prevalence. Multiple linear regression models were used to assess the association between UGS and the SGDS-K scores. Both analyses were adjusted for age, sex, drinking and smoking habits, educational level, marital status, household income, BMI, regular exercise, hypertension, and diabetes mellitus. Age and BMI were treated as continuous variables, while the others were handled as categorical variables.

Subgroup analyses were conducted to examine whether the association between faster UGS and risk of depression differed based on key sociodemographic and health-related factors, including age (<75 vs. ≥75 years), sex (male vs. female), marital status (divorced/widowed/single vs. married/partnered), educational level (≤middle school vs. ≥high school), household income (<3 vs. ≥3 million KRW/month), current drinking habits (yes vs. no), smoking status (never vs. ever), regular exercise (<150 vs. ≥150 min/week), BMI (<25 and ≥25 kg/m$^2$), hypertension status (yes vs. no), and diabetes mellitus status (yes vs. no). To evaluate these associations, the mid group (n=811) from the sex-specific UGS tertile groups was excluded. Furthermore, the low and high UGS groups were compared within each subgroup. The *p*-value for interaction was estimated to assess the consistency of the relationships across the subgroups. All tests were two-tailed, and statistical significance was set at *p*-value <0.05.

## Results

This study included 2,419 participants (1,354 women). Table 1 presents the participants' characteristics, stratified by sex and depression status. Prevalence of depression was higher in women (26.29%) than in men (13.33%) (*p*<0.0001). In both sexes, the depression group had a significantly slower UGS compared with the non-depression group (men: 0.88±0.20 vs. 0.97±0.20 m/s, *p*<0.0001; women: 0.78±0.21 vs. 0.87±0.19 m/s, *p*<0.0001). Furthermore, compared with the non-depression group, the depression group exhibited a significantly higher mean age, SGDS-K score, and higher prevalence of low education level (≤elementary school), low household income (<1 million KRW/month), and individuals living without a partner (divorced, widowed, or single). In men, the depression group had a markedly lower BMI, waist circumference, DBP, and never smokers compared with the non-depression group. In women, the depression group exhibited significantly higher SBP, proportion of hypertension, and individuals who did not engage in regular exercise compared with the non-depression group.

S1 Table presents participants' characteristics based on the sex-specific tertiles of UGS. The high UGS group had a significantly lower prevalence of depression (12.94%) than the low (30.97%) and mid UGS (17.88%) groups (*p*<0.0001). Furthermore, the SGDS-K score was significantly lower in the high UGS group (2.15±2.93) compared with the low (4.15±4.21) and mid UGS (2.68±3.35) groups (*p*<0.0001). Compared with the lower UGS groups, the high UGS group was younger and significantly associated with lower waist circumference and SBP, as well as lower prevalences of living without a partner (divorced, widowed, or single), low education level (≤elementary school) and household income (<1 million KRW/month), hypertension, and diabetes mellitus. However, the high UGS group was associated with higher UGS, T-Chol, TG, and a greater proportion of individuals who engaged in regular exercise compared with the lower UGS groups.

Fig 2 illustrates the age- and sex-stratified comparison of UGS and prevalence of depression. A steady decrease was observed in UGS with aging in both sexes. UGS was significantly faster in men in all age groups than in women, except for the 50–54 age group. Conversely, prevalence of depression steadily increased with aging in both sexes. It was significantly higher in women than in men among those aged ≥55 years.

**Table 1. Participants' characteristics based on depression status and sex.**

| Variables | Men (n = 1,065) | | p-value | Women (n = 1,354) | | p-value |
|---|---|---|---|---|---|---|
| | Non-depression (n = 923) | Depression (n = 142) | | Non-depression (n = 998) | Depression (n = 356) | |
| **Age** (years) | 66.52 ± 7.87 | 70.39 ± 8.26 | <0.0001 | 66.05 ± 8.10 | 69.95 ± 8.18 | <0.0001 |
| **Marital status**, n (%) | | | <0.0001 | | | <0.0001 |
| Divorced/widowed/single | 54 (5.85) | 26 (18.31) | | 256 (25.65) | 132 (37.08) | |
| Married/partnered | 869 (94.15) | 116 (81.69) | | 742 (74.35) | 224 (62.92) | |
| **Education level**, n (%) | | | <0.0001 | | | <0.0001 |
| ≤ Elementary school | 238 (25.78) | 63 (44.37) | | 535 (53.61) | 256 (71.91) | |
| Middle/high school | 630 (68.26) | 72 (50.70) | | 440 (44.09) | 99 (27.81) | |
| ≥ College | 55 (5.96) | 7 (4.93) | | 23 (2.30) | 1 (0.28) | |
| **Household income**, n (%) | | | <0.0001 | | | <0.0001 |
| < 1 (million KRW/month) | 295 (31.96) | 95 (66.90) | | 485 (48.60) | 256 (71.91) | |
| 1–<2 | 247 (26.76) | 26 (18.31) | | 233 (23.35) | 51 (14.33) | |
| 2–<3 | 166 (17.99) | 13 (9.15) | | 116 (11.62) | 22 (6.18) | |
| 3–<4 | 97 (10.51) | 2 (1.41) | | 93 (9.32) | 16 (4.49) | |
| ≥ 4 | 118 (12.78) | 6 (4.23) | | 71 (7.11) | 11 (3.09) | |
| **Drinking habit**, n (%) | | | 0.35 | | | 0.47 |
| Never drinker | 206 (22.32) | 25 (17.60) | | 788 (78.96) | 291 (81.74) | |
| Former drinker | 152 (16.47) | 28 (19.72) | | 24 (2.40) | 9 (2.53) | |
| Current drinker | 565 (61.21) | 89 (62.68) | | 186 (18.64) | 56 (15.73) | |
| **Smoking habit**, n (%) | | | <0.01 | | | 0.50 |
| Never smoker | 237 (25.68) | 19 (13.38) | | 982 (98.40) | 347 (97.47) | |
| Former smoker | 465 (50.38) | 80 (56.34) | | 6 (0.60) | 4 (1.12) | |
| Current smoker | 221 (23.94) | 43 (30.28) | | 10 (1.00) | 5 (1.40) | |
| **Regular exercise**, n (%) | | | 0.11 | | | <0.001 |
| < 150 min/week | 733 (79.41) | 121 (85.21) | | 830 (83.17) | 323 (90.73) | |
| ≥ 150 min/week | 190 (20.59) | 21 (14.79) | | 168 (16.83) | 33 (9.27) | |
| **UGS** (m/s) | 0.97 ± 0.20 | 0.88 ± 0.20 | <0.0001 | 0.87 ± 0.19 | 0.78 ± 0.21 | <0.0001 |
| **SGDS-K score** | 1.11 ± 1.42 | 8.80 ± 2.77 | <0.0001 | 1.70 ± 1.63 | 9.15 ± 2.94 | <0.0001 |
| **BMI** (kg/m²) | 24.17 ± 3.10 | 23.21 ± 3.30 | <0.001 | 24.90 ± 3.36 | 24.74 ± 3.62 | 0.44 |
| **Waist circumference** (cm) | 89.74 ± 8.31 | 88.07 ± 8.95 | <0.05 | 90.23 ± 8.57 | 90.28 ± 9.53 | 0.93 |
| **SBP** (mmHg) | 124.28 ± 15.00 | 125.30 ± 15.74 | 0.46 | 125.26 ± 17.19 | 127.48 ± 17.79 | <0.05 |
| **DBP** (mmHg) | 78.26 ± 9.41 | 76.07 ± 9.67 | <0.05 | 77.43 ± 9.71 | 77.21 ± 10.03 | 0.72 |
| **T-Chol** (mg/dL) | 174.48 ± 33.17 | 175.68 ± 35.90 | 0.69 | 185.09 ± 34.17 | 183.14 ± 32.24 | 0.35 |
| **HDL-C** (mg/dL) | 43.63 ± 12.14 | 44.42 ± 13.96 | 0.52 | 47.38 ± 11.92 | 47.00 ± 11.14 | 0.60 |
| **TG** (mg/dL) | 136.34 ± 87.20 | 143.77 ± 100.41 | 0.40 | 130.03 ± 91.77 | 131.95 ± 69.65 | 0.68 |
| **FBG** (mg/dL) | 103.74 ± 26.70 | 102.72 ± 24.35 | 0.67 | 100.15 ± 25.76 | 99.33 ± 24.37 | 0.60 |
| **Hypertension**, n (%) | 482 (52.22) | 81 (57.04) | 0.28 | 520 (52.10) | 215 (60.39) | <0.01 |
| **Diabetes mellitus**, n (%) | 226 (24.49) | 36 (25.35) | 0.82 | 232 (23.25) | 91 (25.56) | 0.38 |

KRW, Korean won; UGS, usual gait speed; SGDS-K, Korean version of the Geriatric Depression Scale-Short Form; BMI, body mass index; SBP, systolic blood pressure; DBP, diastolic blood pressure; T-Chol, total cholesterol; HDL-C, high-density lipoprotein cholesterol; TG, triglyceride; FBG, fasting blood glucose.

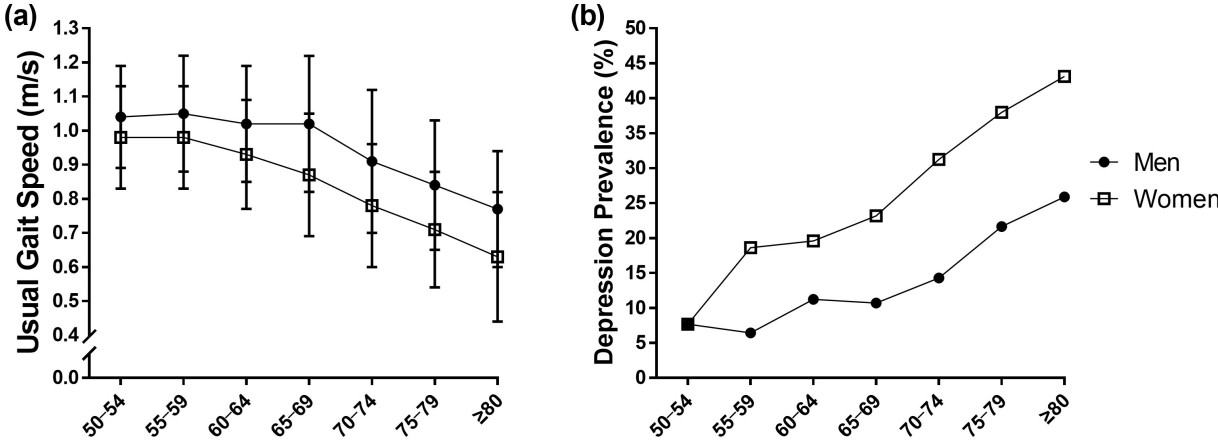

**Fig 2. Comparison of (a) usual gait speed and (b) depression prevalence by age and sex.**

Table 2 presents the association between sex-specific UGS tertiles and risk of depression. Among men, those in the high UGS group had a 50% lower risk of depression compared with participants in the low UGS group, adjusted for covariates ($p < 0.05$; $p$ for trend $<0.05$). Among women, those in the mid and high UGS groups had a 43% ($p < 0.001$) and 44% ($p < 0.01$) lower depression risk, respectively, compared with those in the low UGS group, adjusted for covariates ($p$ for trend $<0.01$).

Table 3 presents the relationship between UGS levels and the SGDS-K scores for each sex. Faster UGS was associated with lower SGDS-K scores in both sexes, adjusted for covariates. The overall regression model was statistically significant in both men and women (all $p < 0.0001$). Specifically, the SGDS-K score was lowered by 0.14 ($p < 0.01$) and 0.33 points ($p < 0.0001$) with each 0.1 m/s increase in UGS among men and women, respectively, indicating a significant linear relationship between UGS and the SGDS-K score across sexes.

**Table 2. Odds ratios for the prevalence of depression according to sex-specific tertiles of UGS.**

| | N | UGS (m/s) | SGDS-K score | Depression, n (%) | Crude Model OR (95% CI) | Model 1 OR (95% CI) | Model 2 OR (95% CI) |
|---|---|---|---|---|---|---|---|
| **Total** | | | | | | | |
| Low UGS | 804 | 0.68±0.13 | 4.15±4.21 | 249 (30.97) | 1 (reference) [c] | 1 (reference) [c] | 1 (reference) [c] |
| Mid UGS | 811 | 0.90±0.07 | 2.68±3.35 | 145 (17.88) | 0.49 (0.38–0.61) [****] | 0.65 (0.50–0.83) [***] | 0.64 (0.49–0.83) [***] |
| High UGS | 804 | 1.12±0.13 | 2.15±2.93 | 104 (12.94) | 0.33 (0.26–0.43) [****] | 0.56 (0.41–0.75) [***] | 0.55 (0.40–0.74) [***] |
| **Men** | | | | | | | |
| Low UGS | 354 | 0.74±0.12 | 2.83±3.57 | 71 (20.06) | 1 (reference) [c] | 1 (reference) [a] | 1 (reference) [a] |
| Mid UGS | 356 | 0.96±0.05 | 2.03±2.92 | 46 (12.92) | 0.59 (0.40–0.89) [*] | 0.82 (0.53–1.28) | 0.83 (0.53–1.29) |
| High UGS | 355 | 1.18±0.11 | 1.54±2.60 | 25 (7.04) | 0.30 (0.19–0.49) [****] | 0.51 (0.30–0.87) [*] | 0.50 (0.29–0.86) [*] |
| **Women** | | | | | | | |
| Low UGS | 450 | 0.63±0.11 | 5.18±4.38 | 178 (39.56) | 1 (reference) [c] | 1 (reference) [b] | 1 (reference) [b] |
| Mid UGS | 455 | 0.85±0.05 | 3.18±3.57 | 99 (21.76) | 0.43 (0.32–0.57) [****] | 0.57 (0.42–0.79) [***] | 0.57 (0.41–0.79) [***] |
| High UGS | 449 | 1.07±0.11 | 2.63±3.08 | 79 (17.59) | 0.33 (0.24–0.44) [****] | 0.56 (0.39–0.82) [**] | 0.56 (0.38–0.82) [**] |

UGS, usual gait speed; SGDS-K, Korean version of the Geriatric Depression Scale-Short Form; OR, odds ratio; CI, confidence interval; BMI, body mass index; [a], $p < 0.05$ in the test for trend of ORs; [b], $p < 0.01$ in the test for trend of ORs; [c], $p < 0.0001$ in the test for trend of ORs; [*], $p < 0.05$; [**], $p < 0.01$; [***], $p < 0.001$; [****], $p < 0.0001$. Model 1 was adjusted for age, sex, drinking, smoking, educational level, marital status, and household income. Model 2 was adjusted BMI, regular exercise, hypertension, and diabetes mellitus in addition to the variables in Model 1.

**Table 3. Associations between per 0.1-unit increment in UGS and the SGDS-K score.**

| Sex | Model | β (95% CI) | p-value |
|---|---|---|---|
| **Total** | Crude model | −0.49008 (−0.55563, −0.42453) | <0.0001 |
| | Model 1 | −0.23841 (−0.31586, −0.16095) | <0.0001 |
| | Model 2 | −0.23812 (−0.31574, −0.16050) | <0.0001 |
| **Men** | Crude model | −0.27863 (−0.36737, −0.18988) | <0.0001 |
| | Model 1 | −0.13777 (−0.23338, −0.04215) | <0.01 |
| | Model 2 | −0.13756 (−0.23339, −0.04174) | <0.01 |
| **Women** | Crude model | −0.54451 (−0.64228, −0.44674) | <0.0001 |
| | Model 1 | −0.33360 (−0.45286, −0.21435) | <0.0001 |
| | Model 2 | −0.33362 (−0.45335, −0.21390) | <0.0001 |

UGS, usual gait speed; SGDS-K, Korean version of the Geriatric Depression Scale-Short Form; BMI, body mass index. Model 1 was adjusted for age, sex, drinking, smoking, educational level, marital status, and household income. Model 2 was adjusted for BMI, regular exercise, hypertension, and diabetes mellitus in addition to the variables in Model 1.

Subgroup analysis was conducted to examine whether the association between faster UGS and reduced depression risk was consistent across various subgroups, including age, sex, marital status, educational level, household income, current drinking habits, smoking status, regular exercise, BMI, hypertension, and diabetes mellitus. Significance of the association between faster UGS and reduced risk of depression differed among some subgroups (S2 Table). This association was statistically significant only in participants aged <75 years ($p < 0.0001$), those who lived with a partner (married or partnered) ($p < 0.001$), had middle school education or lower ($p < 0.01$), low household income (<3 million KRW/month) ($p < 0.0001$), were non-drinkers ($p < 0.01$), did not engage in regular exercise ($p < 0.01$), and had a low BMI (<25 kg/m$^2$) ($p < 0.01$). However, no significant interactions were observed between the associations across all subgroups.

## Discussion

Our findings suggested that a faster UGS was associated with a lower risk of depression in both sexes. Compared with participants with low UGS, men with high UGS had a 50% lower risk of depression and women with mid and high UGS had a 43% and 44% lower risk, respectively. Additionally, the SGDS-K scores were lowered by 0.14 and 0.33 points in men and women, respectively, with each 0.1 m/s increase in UGS. These findings support recommendations to enhance UGS through exercise training to reduce the risk of depression.

Although slower UGS is a clinical indicator of the future risk of cognitive impairment, dementia, mobility disability, CVD, and mortality [8–10], UGS decreases with age during adulthood [11]. Moreover, since slower UGS was significantly associated with loneliness, social isolation [16], and risk factors for depression [17], the association between UGS and risk of depression has garnered increasing attention in Western countries. Recent studies have reported that slower UGS was significantly associated with an increased risk of depression in very old adults aged ≥85 years in Sweden and Finland [22] and older patients with mild cognitive impairment in Turkey [19]. Previous studies have also indicated that an elevated risk of depression was associated with slower UGS in adults aged ≥55 years in England [20] and the Netherlands [21]. However, no study conducted a sex-disaggregated analysis nor considered a crucial confounder, such as whether or not participants met the current exercise guidelines (150 min/week or more of moderate-intensity exercise). A recent study reported that meeting the guidelines was associated with a 33% reduced risk of developing depression among female participants [29]. Hence, further studies should provide stronger evidence for the association between UGS and risk of depression after adjusting for sex as a covariate. In our study population, UGS was significantly higher in men than in women in all the age groups, except for the 50–54 age group. However, the prevalence of depression was higher in

women than in men among those aged ≥55 years, as shown in Fig 2. Considering these sex-based differences, we investigated the risk of depression according to sex-specific UGS tertiles in the Korean population. Our findings indicated that having a faster UGS reduced the risk of depression in both sexes, adjusted for covariates, which included regular exercise and sociodemographic and health-related factors. This result provided evidence that the association reported in previous studies was significant, regardless of sex. However, other cross-sectional studies reported no significant association between UGS and depression risk in older adults (≥60 years) from Iran [30] and older adults aged ≥60 years with mild and moderate dementia from Norway [31]. Considering these contradictory findings, future studies should verify their relationship and deduce causal relationships.

The SGDS-K score is significantly associated with quality of life, well-being, cognitive function, and oxidative stress biomarkers in the Korean older adult population [25,26]. Since these are important predictors for developing depression, verifying whether UGS is related to the SGDS-K score is necessary for prevention. In our study, the SGDS-K scores were lowered by 0.14 and 0.33 points in men and women, respectively, with each 0.1 m/s increase in UGS. This finding indicated that as UGS increased, the degree of depressive symptoms assessed by the SGDS-K score may be proportionally reduced. However, UGS continued to decrease with aging, whereas the prevalence of depression increased in both sexes, as shown in Fig 2. According to previous studies, UGS decreased with age due to shorter step length resulting from reduced muscular strength, balance ability, and ROM of the lower extremities [12–14]. Recent meta-analytical evidence indicated that resistance training (RT) significantly improved UGS; furthermore, addition of balance training (BT) to RT was most effective for enhancing UGS in postmenopausal women [32] and older adults with sarcopenia [33]. Interestingly, no significant improvement was observed in UGS after only protein supplementation without exercise training; in addition, no additive effect was observed when protein supplementation was added to RT or RT combined with BT [32,33]. Our findings and those of recent meta-analyses support recommendations to maintain and/or enhance UGS through regular RT combined with BT to reduce the risk of depression.

Neural plasticity and systemic inflammation may explain the antidepressant effects of faster UGS. First, inadequate signaling by neurotrophins, such as brain-derived neurotrophic factor (BDNF) that promotes neuroplasticity, neuronal survival, and synaptogenesis, could be a potential factor for the development of depression [34]. The BDNF is originally synthesized as a precursor called proBDNF; however, it plays an antagonistic role in neuronal functions. Conversely, proBDNF induces neuronal atrophy and apoptosis and promotes inflammatory cytokines [35,36]. Compared with healthy controls, patients with depression had significantly higher protein and serum levels of proBDNF and its receptor; however, they had low levels of BDNF and its receptor [37]. Since reduced UGS in older adults was significantly related to elevated levels of proBDNF in plasma extracellular vesicles [38], neural plasticity may explain the antidepressant effects of faster UGS. Second, low physical function, which included slower UGS and lower muscular strength, was significantly associated with increased levels of systemic inflammation markers, such as interleukin-6 (IL-6) and C-reactive protein (CRP) [39,40]. In contrast, a recent meta-analysis reported that higher levels of both IL-6 and CRP were significantly associated with the development of depression [41]. Recent meta-analytical evidence also indicated that regular RT, the most effective method to enhance UGS, significantly improved chronic inflammation and circulation of BDNF levels and even reduced depressive symptoms in older adults [42,43]. Hence, maintaining and/or enhancing physical function, including UGS, through regular exercise training is essential.

To our best knowledge, this is the first study to examine the association between UGS, the SGDS-K scores, and risk of depression in middle-aged and older Koreans. However, this study has two limitations. First, this was a cross-sectional study; hence, we were unable to deduce cause-and-effect. Thus, further randomized controlled trials should verify the effectiveness of improving UGS in preventing depression. Second, although the SGDS-K has been validated in the Korean population for screening depression, its actual prevalence may have been either overestimated or underestimated.

## Conclusions

Our study indicated that faster UGS was associated with a lower risk of depression in both sexes. Additionally, the faster the UGS, the lower the SGDS-K scores in both sexes. Therefore, these findings support recommendations to enhance UGS through exercise training to reduce the risk of depression.

## Supporting information

**S1 Table. Participants' characteristics based on sex-specific tertiles of UGS.**
(DOCX)

**S2 Table. Odds ratios for depression prevalence according to sex-specific tertiles of the UGS in the various subgroups.**
(DOCX)

## Author contributions

**Conceptualization:** Jae Ho Park, Hyun-Young Park.

**Data curation:** Hyun-Young Park.

**Formal analysis:** Jae Ho Park.

**Funding acquisition:** Joong-Yeon Lim.

**Investigation:** Jae Ho Park.

**Methodology:** Hyun-Young Park.

**Project administration:** Joong-Yeon Lim.

**Resources:** Joong-Yeon Lim.

**Software:** Jae Ho Park.

**Supervision:** Hyun-Young Park.

**Validation:** Jae Ho Park.

**Visualization:** Jae Ho Park.

**Writing – original draft:** Jae Ho Park.

**Writing – review & editing:** Jae Ho Park, Joong-Yeon Lim, Hyun-Young Park.

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
