## [Decision Letter · Decision Letter 0]

23 Oct 2025

Dear Dr. Park,

Thank you for submitting your manuscript to PLOS ONE. After careful consideration, we feel that it has merit but does not fully meet PLOS ONE’s publication criteria as it currently stands. Therefore, we invite you to submit a revised version of the manuscript that addresses the points raised during the review process.

We look forward to receiving your revised manuscript.

Kind regards,

Masaki Mogi

Academic Editor

PLOS ONE

**Journal Requirements:**

https://www.frontiersin.org/journals/physiology/articles/10.3389/fphys.2022.930922/full

In your revision ensure you cite all your sources (including your own works), and quote or rephrase any duplicated text outside the methods section. Further consideration is dependent on these concerns being addressed.

“This study was supported by the National Institute of Health (NIH) research project (Grant No. 2024-NI-003-01).”

**Additional Editor Comments:**

The manuscript has been well assessed by two reviewers; however, the authors need to revise the present manuscript according to the reviewers' constructive suggestions.

See the comments carefully and respond them appropriately.

Reviewers' comments:

Reviewer's Responses to Questions

**Comments to the Author**

1. Is the manuscript technically sound, and do the data support the conclusions?

Reviewer #1: Yes

Reviewer #2: Partly

2. Has the statistical analysis been performed appropriately and rigorously?

Reviewer #1: No

Reviewer #2: Yes

3. Have the authors made all data underlying the findings in their manuscript fully available?

Reviewer #1: Yes

Reviewer #2: Yes

4. Is the manuscript presented in an intelligible fashion and written in standard English?

Reviewer #1: No

Reviewer #2: Yes

Reviewer #1: This is a South Korean study conducted on 2,419 older adults investigating the association between usual gait speed (UGS) and depression. While this association has already been reported in several previous studies, the originality of this work lies in its geographic setting, the consideration of numerous covariates, and the exploration of potential sex-related differences. However, a few clarifications are needed:

1. Methodological Clarification: In the multivariate models (Table 2), it is unclear how many participants were included in the analyses. According to Table 2, one might assume that the multivariate analyses were conducted on all 2,419 individuals in the cohort. However, considering the large number of covariates included in the models, it seems unlikely that all participants were retained in the multivariate analyses. The exact sample size for each model should be clearly reported. Moreover, the subgroup analyses (Table S2) appear to have been conducted on approximately 1,500 individuals. How were missing data handled? Why were only about 1,500 participants included in the subgroup analyses, particularly those stratified by age?

2. Table 3: Similarly, it is important to specify how many participants were included in Models 1 and 2.

3. Abstract: In the Results section, confidence intervals should be reported alongside effect estimates.

4. Subgroup Analysis (Table S2): The absence of a statistically significant association in the subgroup aged ≤65 years requires further explanation in light of existing literature. Is this lack of significance due solely to insufficient statistical power? Would it be possible to conduct an additional subgroup analysis comparing participants aged <75 years versus those ≥75 years?

5. Discussion: The sentence “Hence, further studies should enhance evidence of the association between UGS and risk of depression after adjusting this as a covariate” is unclear. Are the authors referring to sex or another variable as the covariate to be adjusted for?

6. Discussion – Pathophysiological Mechanisms: The entire section discussing potential pathophysiological mechanisms (lines 285–303) is somewhat confusing and not essential in its current length. It could be summarized in two or three concise sentences to improve clarity and focus.

Reviewer #2: This study investigated the cross-sectional associations of UGS with depression and with the SGDS-K score in a community-based cohort in Korea. I consider that there is room for improvement in the logical flow and clarity of the manuscript. Please refer to the following comments and revise the manuscript.

Major comments:

1) Lines 63-66: The rationale for examining differences in the association between UGS and depression according to sex appears insufficient. In the studies conducted in Western countries, was sex not adjusted for as a confounding factor? It seems that the authors considered sex as a potential effect modifier rather than a confounding factor. Please clarify this point in the manuscript.

2) Lines 71-74: Please elaborate on why the authors examined the associations between UGS and SGDS-K score.

3) Line 81: According to Lines 85-86, in the eighth wave (2015-2016) of KoGES, the largest number of participants had UGS measured. Why did the authors not use a longitudinal design with UGS data in the eighth wave as baseline and later waves to evaluate the onset of depression?

4) Lines 157-159: Please clarify how the covariates were treated (continuous or categorical) and how categorical variables were categorized.

Minor comments:

1) Line 61: “an inverse association between slower UGS and risk of depression” may be inappropriate. “inverse” or “slower” might be unnecessary.

2) Line 81: Please state the study design (i.e., cross-sectional study) in the materials and methods section.

3) Lines 93-95: Please report the number of participants in the results section rather than the materials and methods section.

4) Lines 118-142: Many covariates are listed, but some, such as laboratory values, do not appear to have been used in evaluating the association between UGS and depression or SGDS-K score. Consider including only relevant variables.

5) Lines 170-191: Please highlight the differences in UGS by depression status, and in depression and SGDS-K score by the UGS group.

6) Lines 200, 204, 234, 235: It seems more appropriate to use “odds” rather than “risk”.

7) Table 2: It may be more appropriate to present the SGDS-K score using the median and IQR rather than mean ± SD. If this change is made, please also revise the description in Line 147 accordingly. In addition, P values should be reported numerically rather than indicated with asterisks.

8) Lines 210-213: Please show that UGS and SGDS-K score has a liner relationship.

9) Lines 225, 233, Conclusions: As this study is cross-sectional and cannot adequately establish causality, “protective benefit” appears to be overstated. Please revise the wording.

10) There are several redundant statements in the manuscript, such as the repeated mention that SGDS-K is a valid instrument. Please carefully review the entire manuscript and reduce redundancy where appropriate, including those not mentioned here.

**Do you want your identity to be public for this peer review?** For information about this choice, including consent withdrawal, please see our Privacy Policy

Reviewer #1: **Yes:** Denis Boucaud-Maitre

Reviewer #2: No

---

## [Author Response · Author response to Decision Letter 1]

17 Nov 2025

Response to Reviewer 1 Comments

Manuscript Number: PONE-D-25-35976

Title: Usual gait speed is inversely associated with depression in middle-aged and older adults: A cross-sectional study in Korea

We would like to thank you for your positive review and encouraging comments regarding our study. We have responded to each point below and indicated the corresponding changes in the revised manuscript in red. Additionally, we have included the modified text in quotation marks below each comment.

Point 1: 1. Methodological Clarification: In the multivariate models (Table 2), it is unclear how many participants were included in the analyses. According to Table 2, one might assume that the multivariate analyses were conducted on all 2,419 individuals in the cohort. However, considering the large number of covariates included in the models, it seems unlikely that all participants were retained in the multivariate analyses. The exact sample size for each model should be clearly reported. Moreover, the subgroup analyses (Table S2) appear to have been conducted on approximately 1,500 individuals. How were missing data handled? Why were only about 1,500 participants included in the subgroup analyses, particularly those stratified by age?

Response 1: Thank you for your careful review of our manuscript. First, as shown in Fig. 1, participants with missing data on any covariates were excluded prior to the analyses. Therefore, the same 2,419 participants were included in all the models (crude model, model 1, and model 2) presented in Table 2. Second, subgroup analyses were conducted after the mid group (n=811), from the three sex-specific UGS tertile groups (low, mid, and high), was excluded. Comparisons were made between the low and high UGS groups. Accordingly, we have added a sentence in the Results section (page 8, lines 173–176) to clarify why only approximately 1,500 participants were included in the subgroups.

"To evaluate these associations, the mid group (n=811) from the sex-specific UGS tertile groups was excluded. Furthermore, the low and high UGS groups were compared within each subgroup."

Point 2: Table 3: Similarly, it is important to specify how many participants were included in Models 1 and 2.

Response 2: Thank you for your comment. As shown in Fig. 1, participants with missing data on any covariates were excluded prior to the analyses. Therefore, the same 2,419 participants were included in all models (crude model, model 1, and model 2) presented in Table 3.

Point 3: Abstract: In the Results section, confidence intervals should be reported alongside effect estimates.

Response 3: We thank you for your valuable comment. We have reported the odds ratios (ORs) along with their 95% confidence intervals (CIs) in the results of the Abstract (page 2, lines 31–34). In addition, we explicitly described their estimation in the Methods (page 2, lines 25–28).

"Multiple linear and logistic regression models were used to assess the association between UGS and SGDS-K scores and estimate the odds ratios (ORs) with 95% confidence intervals (CIs) for the risk of depression, respectively."

Point 4: Subgroup Analysis (Table S2): The absence of a statistically significant association in the subgroup aged ≤65 years requires further explanation in light of existing literature. Is this lack of significance due solely to insufficient statistical power? Would it be possible to conduct an additional subgroup analysis comparing participants aged <75 years versus those ≥75 years?

Response 4: We agree that the absence of statistical significance may be attributable to insufficient statistical power. Therefore, we conducted a sensitivity analysis including the "<75 vs. ≥75 years" subgroup classification suggested by the reviewer. The results are provided in the Response to Reviewer Comments file. We also revised the Methods section (page 8, line 169), the Results section (page 15, lines 245–249), and S2 Table to reflect this age subgroup classification. As shown in the table, although the result was not statistically significant due to the small sample size (n=678) in the <65 years group, the odds ratio was 0.62, suggesting potential benefits of higher UGS. Importantly, no significant interaction was observed in the subgroup analysis based on the 65 years cutoff (p for interaction=0.31), indicating no difference between age groups. Meanwhile, the prevalence of depression remained high in the ≥75 years group compared to other age groups, even among those with high UGS (26%). Furthermore, although not statistically significant, the benefits of high UGS appeared lowest among all age groups, with an OR of 0.80 (indicating 20% lower odds of depression). This finding may imply that other age-related factors influenced the association between UGS and depression. However, as only 415 participants were included in this age group, additional analyses could not be performed. Similarly, the subgroup analysis based on the 75 years cutoff showed no significant interaction (p for interaction=0.28), again indicating no difference between the two groups. In line with the reviewer’s suggestion, we aim to conduct a follow-up study using a larger cohort to further clarify the association between UGS and depression in older adults.

Point 5: Discussion: The sentence “Hence, further studies should enhance evidence of the association between UGS and risk of depression after adjusting this as a covariate” is unclear. Are the authors referring to sex or another variable as the covariate to be adjusted for?

Response 5: We sincerely appreciate the reviewer’s careful reading and for pointing out this ambiguity. Sex is the variable to be adjusted for as a covariate. Hence, we have revised the sentence in the Discussion section (page 16, lines 272–273) accordingly.

"Hence, further studies should provide stronger evidence for the association between UGS and risk of depression after adjusting for sex as a covariate."

Point 6: Discussion – Pathophysiological Mechanisms: The entire section discussing potential pathophysiological mechanisms (lines 285–303) is somewhat confusing and not essential in its current length. It could be summarized in two or three concise sentences to improve clarity and focus.

Response 6: We appreciate your feedback. We believe that this paragraph provides important background information on the potential pathophysiological mechanisms that may underlie the association between UGS and depression. This discussion helps interpret our findings and may inspire future research directions. Therefore, with the reviewer’s kind permission, we would like to retain this section in its current form.

Response to Reviewer 2 Comments

Manuscript Number: PONE-D-25-35976

Title: Usual gait speed is inversely associated with depression in middle-aged and older adults: A cross-sectional study in Korea

We thank you for your positive review and encouraging comments regarding our manuscript. We have responded to each of your raised points below and indicated the corresponding changes in the revised manuscript in red. Additionally, we have included the modified text in quotation marks below each response.

Point 1: Lines 63-66: The rationale for examining differences in the association between UGS and depression according to sex appears insufficient. In the studies conducted in Western countries, was sex not adjusted for as a confounding factor? It seems that the authors considered sex as a potential effect modifier rather than a confounding factor. Please clarify this point in the manuscript.

Response 1: Thank you for your careful review of our manuscript. We thoroughly examined the relevant studies conducted in Western countries and found that although sex was adjusted as a confounding variable, stratified analyses were not performed. Furthermore, most studies acknowledged this limitation. We have added this information to the Introduction section (page 4, lines 65–66).

"; however, they did not examine sex-based differences through stratified analyses."

Second, these studies have reported a higher proportion of women in the slow UGS group. Furthermore, previous research has indicated that the pattern of UGS changes in adulthood shows a gradual decline in men, whereas women experience a marked decrease at menopause (Park et al., 2023). Therefore, considering these sex differences, sex-disaggregated analyses are warranted. We have also incorporated these sentences into the Introduction (page 4, lines 66–69). We expect this addition to naturally lead to the subsequent sentence emphasizing the need for studies targeting Asian populations, thereby strengthening the rationale for our study.

"A study reported that UGS changes exhibit a gradual decline in men in adulthood and a marked decrease at menopause among women (Park et al., 2023); furthermore, previous studies have reported a significantly higher proportion of women in slow UGS groups. Thus, sex-disaggregated analyses are warranted. "

Point 2: Lines 71-74: Please elaborate on why the authors examined the associations between UGS and SGDS-K score.

Response 2: We appreciate the reviewer’s insightful comment regarding the rationale for examining the associations between UGS and the SGDS-K score. We conducted these analyses as the SGDS-K, also reflects a broader spectrum of psychological and physiological health conditions, beyond just serving as a validated screening tool for depression. Previous studies have reported that higher SGDS-K scores were associated with lower well-being, poorer quality of life, cognitive decline, and increased oxidative stress. These findings indicate that the SGDS-K identifies depressive symptoms and represents the underlying biological and psychosocial mechanisms involved in its development. We have added a detailed explanation in the revised Discussion section (page 4, lines 76–79).

"Taken together, these findings suggest that the SGDS-K score functions as a diagnostic instrument for depression and an independent measure directly associated with the key determinants of its development. Therefore, we aimed to investigate whether UGS was significantly associated with the risk of depression and SGDS-K score."

Point 3: Line 81: According to Lines 85-86, in the eighth wave (2015-2016) of KoGES, the largest number of participants had UGS measured. Why did the authors not use a longitudinal design with UGS data in the eighth wave as baseline and later waves to evaluate the onset of depression?

Response 3: We agree with your suggestion. However, if we set the eighth wave (2015–2016) of the KoGES Anseong cohort as the baseline for a longitudinal design, participants with pre-existing depression at this baseline must be excluded. This exclusion substantially reduces the number of eligible participants. Additionally, excluding those lost to follow-up in subsequent waves further decreases the sample size, making it difficult to ensure sufficient statistical power. Therefore, we chose a cross-sectional design. Nevertheless, as recommended by the reviewer, we plan to conduct follow-up longitudinal analyses using larger cohorts.

Point 4: Lines 157-159: Please clarify how the covariates were treated (continuous or categorical) and how categorical variables were categorized.

Response 4: Thank you for your valuable comment. We clarified that age and BMI were treated as continuous variables, while other covariates were handled as categorical variables. Specific categorization has been described in the Covariates section (pages 6–7, lines 125–148) of the Materials and Methods. We have also added the following sentence to the Statistical Analysis section (page 8, lines 165–166).

"Age and BMI were treated as continuous variables, while the others were handled as categorical variables. "

Point 5: Line 61: “an inverse association between slower UGS and risk of depression” may be inappropriate. “inverse” or “slower” might be unnecessary.

Response 5: We appreciate your feedback. We agree that the expression “inverse association between slower UGS and risk of depression” could be confusing. We revised the sentence in the Introduction section (page 3, lines 62–64) to improve clarity and readability.

"Several studies have reported that slower UGS is associated with an increased risk of depression in Western countries, …”

Point 6: Line 81: Please state the study design (i.e., cross-sectional study) in the materials and methods section.

Response 6: In accordance with the reviewer’s comment, we have explicitly stated that this study is a cross-sectional study in the Materials and Methods section (page 5, lines 87–89).

"This population-based cross-sectional study used data from the eighth wave (2015–2016) of the Anseong cohort, part of the Korean Genome and Epidemiology Study (KoGES) that aims to establish comprehensive healthcare guidelines for non-communicable diseases."

Point 7: Lines 93-95: Please report the number of participants in the results section rather than the materials and methods section.

Response 7: Thank you for your comment. We agree that presenting the number of participants in the Results would enhance clarity. The inclusion and exclusion criteria of participants are generally described in the “Study Participants” subsection of the Materials and Methods section. Nevertheless, to address the reviewer’s concern and improve the transparency of our reporting, we have additionally provided the number of participants in the Results section (page 9, line 181).

"This study included 2,419 participants (1,354 women)."

Point 8: Lines 118-142: Many covariates are listed, but some, such as laboratory values, do not appear to have been used in evaluating the association between UGS and depression or SGDS-K score. Consider including only relevant variables.

Response 8: We sincerely appreciate the reviewer’s valuable comment. To provide comprehensive baseline information of the participants, we have presented laboratory test values measured for nearly all participants in the KoGES Anseong cohort. However, as noted, only variables with a demonstrated association with UGS and depression or the SGDS-K score were used as covariates in the adjusted analyses. However, we respectfully ask for your understanding and permission to include the basic laboratory test values, as this additional information may be of interest and benefit to readers.

Point 9: Lines 170-191: Please highlight the differences in UGS by depression status, and in depression and SGDS-K score by the UGS group.

Response 9: Thank you for your comment. Following the reviewer's suggestion, we have added a sentence to the Results section (page 9, lines 183–185) comparing UGS (m/s) according to depression status.

"In both sexes, the depression group had a significantly slower UGS compared with the non-depression group (men: 0.88 ± 0.20 vs. 0.97 ± 0.20 m/s, p<0.0001; women: 0.78 ± 0.21 vs. 0.87 ± 0.19 m/s, p<0.0001)."

Second, we have also included a sentence in the Results section (page 11, lines 200–203) comparing the prevalence of depression and SGDS-K scores among different UGS groups.

"The high UGS group had a significantly lower prevalence of depression (12.94%) than the low (30.97%) and mid UGS (17.88%) groups (p<0.0001). Furthermore, the SGDS-K score was significantly lower in the high UGS group (2.15 ± 2.93) compared with the low (4.15 ± 4.21) and mid UGS (2.68 ± 3.35) groups (p<0.0001)."

Point 10: Lines 200, 204, 234, 235: It seems more appropriate to use “odds” rather than “risk”.

Response 10: Thank you for your valuable suggestion regarding the use of “odds” instead of “risk.” We understand that odds ratios are generally preferred in cross-sectional studies. However, to our knowledge, the use of the term “risk” in studies reporting odds ratios is also acceptable. With your permission, we would like to continue using “risk” in the manuscript. We appreciate your consideration.

Point 11: Table 2: It may be more appropriate to present the SGDS-K score using the median and IQR rather than mean ± SD. If this change is made, please also revise the description in Line 147 accordingly. In addition, P values should be reported numerically rather than indicated with asteri

---

## [Decision Letter · Decision Letter 1]

23 Nov 2025

Usual gait speed is inversely associated with depression in middle-aged and older adults: A cross-sectional study in Korea

PONE-D-25-35976R1

Dear Dr. Park,

We’re pleased to inform you that your manuscript has been judged scientifically suitable for publication and will be formally accepted for publication once it meets all outstanding technical requirements.

Kind regards,

Masaki Mogi

Academic Editor

PLOS ONE

Additional Editor Comments (optional):

Reviewers' comments:

Reviewer's Responses to Questions

**Comments to the Author**

Reviewer #1: All comments have been addressed

Reviewer #2: (No Response)

2. Is the manuscript technically sound, and do the data support the conclusions?

Reviewer #1: Yes

Reviewer #2: (No Response)

3. Has the statistical analysis been performed appropriately and rigorously?

Reviewer #1: Yes

Reviewer #2: (No Response)

4. Have the authors made all data underlying the findings in their manuscript fully available?

Reviewer #1: Yes

Reviewer #2: (No Response)

5. Is the manuscript presented in an intelligible fashion and written in standard English?

Reviewer #1: Yes

Reviewer #2: (No Response)

Reviewer #1: The authors have adequately addressed all of my comments. From my perspective, the manuscript is acceptable.

Reviewer #2: The authors have addressed most of my comments satisfactorily, and I appreciate their efforts to revise the manuscript.

**Do you want your identity to be public for this peer review?** For information about this choice, including consent withdrawal, please see our Privacy Policy

Reviewer #1: **Yes:** Denis Boucaud-Maitre

Reviewer #2: No

---

## [Editor Report · Acceptance letter]

PONE-D-25-35976R1

PLOS One

Dear Dr. Park,

I'm pleased to inform you that your manuscript has been deemed suitable for publication in PLOS One. Congratulations! Your manuscript is now being handed over to our production team.

Kind regards,

on behalf of

Dr. Masaki Mogi

Academic Editor

PLOS One